# HLbL in muon g-2 at large loop momenta

Johan Bijnens[1], Nils Hermansson-Truedsson [2] and Antonio Rodríguez-Sánchez[3]*

**1** Department of Astronomy and Theoretical Physics, Lund University, Box 43, SE221-00 Lund, Sweden
**2** Albert Einstein Center for Fundamental Physics, Institute for Theoretical Physics, Universität Bern, Sidlerstrasse 5, CH–3012 Bern, Switzerland
**3** Université Paris-Saclay, CNRS/IN2P3, IJCLab, 91405 Orsay, France
* arodriguez@ijclab.in2p3.fr

December 17, 2021

## Abstract

We study the HLbL contribution to g-2 in the kinematic region where the three loop momenta are large. We show how, even when the fourth photon is in the static limit, the massless quark loop gives the leading term of an operator product expansion. Power corrections are found to be small. Gluonic corrections are also included and the expansion is found to be well-behaved at relatively low-energies, which can be used to reduce uncertainties in the HLbL contribution to g-2.

## 1 Introduction

A new measurement of the anomalous magnetic moment of the muon, coming from the FNAL Muon g-2 Experiment, was released this year [1], updating the combined experimental average

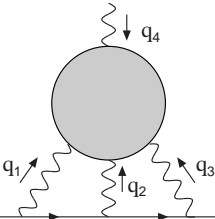

Figure 1: HLbL contribution to g-2. Figure reused from [5].

to

$$a_\mu^{\text{exp}} = 11659\,2061(41) \cdot 10^{-11}\,, \tag{1}$$

and confirming the tension with the SM prediction of Ref. [2],

$$a_\mu^{\text{SM}} = 11659\,1810(43) \cdot 10^{-11}\,. \tag{2}$$

Such a remarkable theoretical precision is in part a consequence of the strong dominance of pure QED contributions, whose theory uncertainties are completely negligible compared to the experimental result. Electroweak corrections are very small and well known too, so they cannot, within the SM, account for such a tension. A discrepancy of $\sim 2.5 \cdot 10^{-8}$ with perfectly controlled corrections from strong interactions would immediately translate into evidence of BSM physics. In order to achieve such a control, strong efforts are being made by both the data-driven and lattice approaches in the two relevant contributions: Hadronic Vacuum Polarization (HVP) and Hadronic Light-by-Light contributions (HLbL). The focus of our work, [3–5], has been oriented to improve the data-driven evaluation of the latter.

## 2 Hadronic Light-by-Light contributions to the muon g-2

The Hadronic Light-by-Light contribution to g-2 is depicted in Fig. 1. The key piece for its calculation is the corresponding HLbL tensor, defined as

$$\Pi^{\mu_1\mu_2\mu_3\mu_4} \equiv -i \int \frac{d^4 q_4}{(2\pi)^4} \left( \prod_{i=1}^{4} \int d^4 x_i\, e^{-iq_i x_i} \right) \langle 0| T \left( \prod_{j=1}^{4} J^{\mu_j}(x_j) \right) |0\rangle\,, \tag{3}$$

where $J^\mu(x) = \bar{q}(x) Q_q \gamma^\mu q(x)$ is the electromagnetic quark current. The correponding g-2 number is obtained when convoluting

$$\lim_{q_4 \to 0} \frac{\partial \Pi^{\mu_1\mu_2\mu_3\nu_4}}{\partial q_4^{\mu_4}}\,, \tag{4}$$

with the other two loops [6, 7]

$$a_\mu^{\text{HLbL}} = \frac{2\alpha^3}{3\pi^2} \int_0^\infty dQ_1 \int_0^\infty dQ_2 \int_{-1}^{1} d\tau\, \sqrt{1-\tau^2}\, Q_1^3 Q_2^3 \sum_{i=1}^{12} T_i(Q_1, Q_2, \tau) \overline{\Pi}_i(Q_1, Q_2, \tau)\,. \tag{5}$$

The $\overline{\Pi}_i$ functions only depend on the five independent scalar functions of the Lorentz decomposition of Eq. (4) [5]. The main difficulty in the evaluation of this contribution is finding those functions. The weights $T_i$ enhance the low-energy contributions, since the scale of the problem is the muon mass. Perturbative QCD cannot be used for the tensor of Eq. (3) at that energy and non-perturbative methods have to be applied. QCD-inspired evaluations of it gave a first assessment of this contribution [8–10], but a more model-independent evaluation of

it was desirable. That possibility arrived with the data-driven approach of Ref. [6], where a model-independent way of evaluating the different leading long-distance contributions was assessed.[1]

While systematic improvements on them are possible, the residual contributions from medium and short distances are more challenging. One of the expected contributions at short distances was the quark loop, but the low-energy scale of the muon mass and the static g-2 photon requires some lower cut-off, since the logarithmic mass divergences associated to the light-by-light contributions do not make much sense for light-quarks. There are no physical hadronic states with such small masses. A constituent quark mass was typically used as a regulator. However, in the context of a data-driven evaluation, such a model-dependent solution was not satisfactory. Our work has been focused on the interplay of the short-distance contributions to g-2 within QCD, aiming to see whether and how the quark loop is the leading term of any asymptotic expansion in the g-2 kinematics.

## 3  The Operator Product Expansion in the QCD vacuum

Two-point correlation functions of quark currents $J_1$ and $J_2$,

$$\Pi(q) = \int d^4x \, e^{-iqx} \langle 0|T(J_1(x)J_2(0))|0\rangle \,, \tag{6}$$

are described at large Euclidean Momenta by its Operator Product Expansion (OPE) in the presence of the QCD vacuum [12],

$$\Pi \approx \sum_{i,D} \frac{c_{i,D}\mathcal{O}_{i,D}}{Q^D} \,, \tag{7}$$

where $D$ is the dimension of the associated operator $\mathcal{O}_{i,D}$, $c_{i,D}$ are $c$-numbers, the Wilson coefficients, and $Q^2 = -q^2$. Any possible operator with the same quantum numbers as the QCD vacuum can give a nonzero contribution, starting by unity, which provides the perturbative series, and supplemented by operators with quarks and gluons such as the scalar current $\bar{q}q$, which gives rise to the well-known quark condensate, $\langle\bar{q}q\rangle$. A known limitation of this description for some of these expansions is in the separation of the tail of the perturbative expansion, of asymptotic nature, and the effect of vacuum condensates. Some possible prescriptions on how to separate those effects can be found in the literature (e.g. see [13]). The same OPE can be applied to correlation functions with a higher number of external legs. A detailed study of the OPE of three-point correlation functions can be found in Ref. [14].

We then have an asymptotic expansion describing the behaviour of the HLbL tensor of Eq. (3) at large Euclidean momenta, whose leading contribution is clearly given by the quark loop. Its applicability to the muon g-2 kinematics is, however, not obvious even when we explore the region with three large (Euclidean, $Q_{1,2,3}$) loop momenta. This is because the fourth (g-2) momentum is defined in the static $q_4 \rightarrow 0$ limit, where the use of such a vacuum OPE is not justified. Indeed, an explicit exploration shows how the first dimensional correction, the one associated to the quark condensate, presents divergences in that limit from the diagram shown in Fig. 2 [3]. This observation may cast some doubts on the validity of the quark loop as the leading term of the (subleading) short-distance contributions to HLbL g-2.

---

[1]Very recently lattice methods have achieved a similar precision [11]. The very good agreement between two largely model-independent and completely different approaches reinforces the idea that a miss-evaluation of HLbL cannot be behind the current tension in muon g-2.

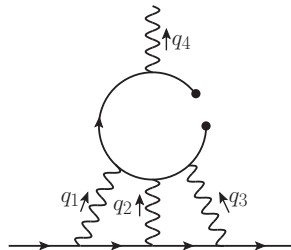

Figure 2: Quark condensate contribution to HLbL. In the static $q_4 \to 0$ limit the quark propagator on the left diverges. Figure reused from [3].

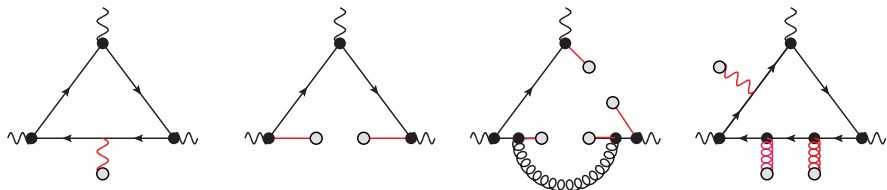

Figure 3: Topologies giving the different contributions to the studied OPE.

## 4 The Operator Product Expansion in the QCD vacuum with an external background photon field

Analogous problems for extrapolating the QCD vacuum OPE to the static-momentum regime were found when trying to describe the nucleon magnetic moments through sum rules. A very elegant solution was presented in Refs. [15, 16], and some applications to other muon contributions can actually be found in the literature, e.g. [17]. The essential point is that the OPE should be performed keeping the external g-2 photon as an explicit external (background) state. This is applying the OPE to the (equivalent) tensor

$$\Pi^{\mu_1\mu_2\mu_3}(q_1, q_2) \equiv -\frac{1}{e} \int \frac{d^4 q_3}{(2\pi)^4} \left( \prod_{i=1}^{3} \int d^4 x_i \, e^{-iq_i x_i} \right) \langle 0| T \left( \prod_{j=1}^{3} J^{\mu_j}(x_j) \right) |\gamma_E(q_4)\rangle, \quad (8)$$

where the $\gamma_E(q_4)$ is the external static photon. Essentially, from such an OPE one needs to keep not those local operators with the same quantum numbers as the QCD vacuum, but those operators with the same quantum numbers as the external g-2 photon, $F_{\mu\nu}$,

$$Q_{1,\mu\nu} \equiv e \, e_q F_{\mu\nu}, \quad Q_{2,\mu\nu} \equiv \bar{q} \sigma_{\mu\nu} q \quad , ... \quad (9)$$

The leading contribution is associated to the lowest dimensional operator, $Q_1$. This is represented in the first diagram from the left of Fig. 3. We showed in Ref. [3] how, up to gluonic corrections, it corresponds to the massless quark loop, which becomes the leading term in a well-defined expansion. In the neighbourhood of the $m_q \to 0$ limit, the leading quark mass correction is not quadratic but linear in the mass and is given by the contribution of the second diagram in Fig 3 to the tensor current, which has a non-zero expectation value in the presence of an external background photon even in the strict chiral limit. This effect of spontaneous chiral symmetry breaking is known as the magnetic susceptibility of the QCD vacuum and is trivially related to the zero momentum limit of $\Pi_{VT}(q)$. The explicit results associated to these contributions, which scale as

$$\frac{(4\pi)^2 m_q X_q}{Q^2} \sim \frac{\Lambda_\chi^2}{M_\rho^2} \frac{M_{\pi(K)}^2}{Q^2}, \quad (10)$$

can be found in Refs. [3,4] and may be used as short-distance constraints for possible evaluations of mass corrections to chiral limit evaluations of HLbL.

Beyond the chiral limit, the leading power corrections are suppressed by four powers of the large loop momenta. The associated topologies are the ones of Fig. 3. Details in the computation and explicit expressions for them can be found in Ref. [4]. We have checked that, as far as one is above 1 GeV, the power corrections are typically suppressed by at least two orders of magnitude with respect to the leading massless quark loop and can be safely neglected.

## 5 The two-loop corrections

In principle the gluonic corrections to the HLbL tensor contain many different two-loop diagrams which depend on where to connect the gluon lines, also with respect to the soft static photon. However, a simplification occurs when realizing that the color structure carried by the gluon can be factored out and it can be regarded as an extra "photon". Then, before contracting the gluon propagator and setting the external g-2 momentum to zero, we simply have a fully symmetric sum of hexagons. It is then a natural step to take advantage of this strong symmetry to eventually break it with both the soft-photon and the second (gluon-propagator) loop. This very simple set-up leads to a very large number of two-loop integrals. But then we can use general projectors $P_{\mu_1\mu_2\mu_3\mu_4\nu_4}^{\tilde{\Pi}_i}$ to the five independent scalar functions $\tilde{\Pi}_i$ of the HLbL g-2 tensor. One finds [5]

$$
\begin{aligned}
\tilde{\Pi}_i &= P_{\mu_1\mu_2\mu_3\mu_4\nu_4}^{\tilde{\Pi}_i} \lim_{q_4\to 0} \frac{\partial \Pi^{\mu_1\mu_2\mu_3\nu_4}}{\partial q_4^{\mu_4}} \\
&= -\frac{(N_c^2-1)g_s^2 e_q^4}{4} \int \frac{d^4q_5}{(2\pi)^4} \frac{g_{\mu_5\mu_6}}{q_5^2} \lim_{\substack{q_4\to 0 \\ q_6\to -q_5}} P_{\mu_1\mu_2\mu_3\mu_4\nu_4}^{\tilde{\Pi}_i} \frac{\partial}{\partial q_4^{\nu_4}} H^{\mu_1\mu_2\mu_3\mu_4\mu_5\mu_6},
\end{aligned}
\tag{11}
$$

where

$$
\begin{aligned}
H^{\mu_1\mu_2\mu_3\mu_4\mu_5\mu_6} \equiv \int \frac{d^4p}{(2\pi)^4} \sum_{\sigma(1,2,4,5,6)} \mathrm{Tr}\Big( &\gamma^{\mu_3} S(p+q_1+q_2+q_4+q_5+q_6)\gamma^{\mu_1} S(p+q_2+q_4+q_5+q_6) \\
&\times \gamma^{\mu_2} S(p+q_4+q_5+q_6)\gamma^{\mu_4} S(p+q_5+q_6)\gamma^{\mu_5} S(p+q_6)\gamma^{\mu_6} S(p)\Big).
\end{aligned}
\tag{12}
$$

$S(p) = \frac{\not p}{p^2}$ is the massless quark propagator and $\sigma(1,2,4,5,6)$ the set of pairwise permutations of $\mu_i$ and $q_i$ for $i = 1, 2, 3, 5, 6$.

Now we "simply" need to deal with $\sim 10^{3,4}$ different scalar two-loop integrals, depending on the set of used projectors (which are related by gauge invariance). In order to get the task done we take advantage of KIRA [18], which is a software that allows for a reduction of multi-loop scalar integrals into a set of master integrals. We find that there is a set of them, corresponding to the integrals of Fig 4, for which all the needed $\frac{1}{D-4}$ expansions have been worked out analytically in terms of classical poly-logs [19].

We then have fully analytic results for the five independent scalar functions of the HLbL tensor encoding the g-2 kinematics, which are valid for the regime of large momenta, including the leading term, power and gluonic corrections. We find that gluonic corrections are typically suppressed with respect to the quark loop by a factor of approximately $-\frac{\alpha_s}{\pi}$.[2]

---

[2]Before realizing of a typo in one of the analytic expressions for the master integrals presented in Ref. [19], the obtained prefactor in front of $\frac{\alpha_s}{\pi}$ was $\mathcal{O}(10^{2,3})$, which would have completely killed the applicability of this expansion.

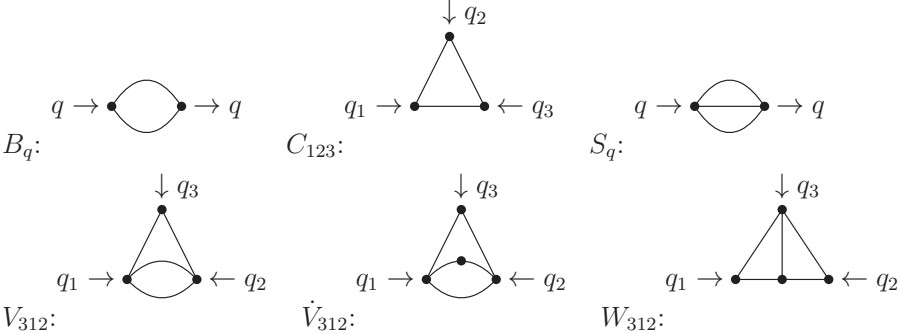

Figure 4: Master integrals entering into the two-loop calculation after applying scalar reduction. Figure reused from [5].

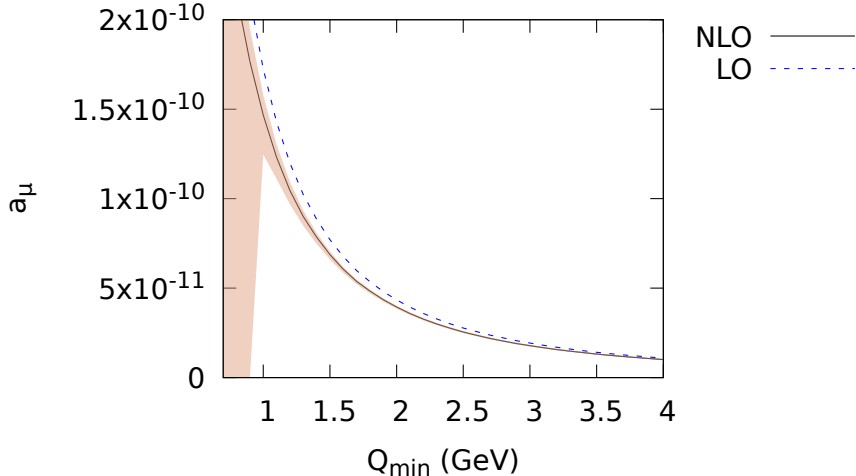

Figure 5: Fully short-distance contributions for HLbL muon g-2, as obtained by our expansion. The leading term (dashed-line) corresponds to the massless quark loop and the NLO one includes the gluonic corrections. Figure reused from [5].

Now it is a matter of convoluting with the other two loops and numerically integrate gluonic corrections, using Eq. (5), for the kinematics regions where the expansion is valid. This is $Q_{1,2,3} > Q_{\min}$. Having computed the power and gluonic corrections, we can actually assess from which values of $Q_{\min}$ the expansion is safe. The result for the leading massless quark loop and the gluonic corrections is displayed in Fig. 5. We observe how, as far as the running of the strong coupling is well defined, the gluonic corrections are rather small and negative. As a consequence, the obtained expansion gives a very precise description of the (subleading, but not negligible) fully short-distance contributions of HLbL muon g-2.

In order to improve the numerical precision, explicit analytic expansions have been performed in some kinematic regions to cancel spurious singularities. One of them corresponds to a sub-region of the Melnikov-Vainshtein short-distance constraints [20]. We have checked how we recover the correct limits for both the quark loop and the gluonic corrections [21,22].

# 6 Conclusions

We have shown how, in spite of the soft-momentum associated to the external g-2 photon, an OPE can be applied for the HLbL muon g-2 contribution when the three loop momenta are large. The leading contribution is given by the massless quark loop [3].

The leading power correction is found to be suppressed by two powers of the energy, from which one corresponds to the quark mass. In the chiral limit, the first power corrections appear suppressed by four powers of the energy. We have worked out explicit expressions for them in Ref. [4]. They are found to be negligible as far as one is above $\sim 1\,\text{GeV}$.

Finally, we have computed the gluonic corrections. The relatively small prefactors in the $\frac{\alpha_s}{\pi}$ expansion confirms that it is well behaved from relatively low energies and can be used to improve the precision of the data-driven evaluation of muon g-2.

# Acknowledgements

We thank Laetitia Laub for a fruitful and enjoyable collaboration. This research is supported in part by the Albert Einstein Center for Fundamental Physics at Universität Bern (NHT), the Swedish Research Council grants contract numbers 2016-05996 and 2019-03779 (JB) and the Agence Nationale de la Recherche (ANR) under grant ANR-19-CE31-0012 (project MORA) (ARS).

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
