# Peer review of "HLbL in muon g-2 at large loop momenta"

_SciPost Physics Proceedings_

## Round 1 · Referee Report · Swagato Banerjee (Referee 1) · 2024-11-30

Report

Data-driven techniques are used to improve upon the evaluation of hadronic light-by-light contribution to the anomalous magnetic moment of the muon. Although the momentum associated to the external photon is soft, operator product expansion can be applied for the hadronic light-by-light muon g-2 contribution when the three loop momenta are large, and the leading contribution is given by the massless quark loop. The leading power correction is found to be suppressed by two powers of the energy, from which one corresponds to the quark mass. In the chiral limit, the first power corrections appear suppressed by four powers of the energy. The gluonic corrections are found to be well-behaved at relatively low energies and can be used to
improve the precision of the data-driven evaluation of muon g-2.

Recommendation

Publish (surpasses expectations and criteria for this Journal; among top 10%)

---

## Editorial Decision

accepted_in_target_journal